# Molecular Characterization of a Novel Ourmia-Like Virus Infecting *Phoma matteucciicola*

**DOI:** 10.3390/v12020231

**Published:** 2020-02-19

**Authors:** Jia Zhou, Yuhua Wang, Xiaofei Liang, Changping Xie, Wenbo Liu, Weiguo Miao, Zhensheng Kang, Li Zheng

**Affiliations:** 1Key Laboratory of Green Prevention and Control of Tropical Plant Disease and Pests, Ministry of Education and College of Plant Protection, Hainan University, Haikou 570228, China; 2State Key Laboratory of Crop Stress Biology for Arid Areas and College of Plant Protection, Northwest A&F University, Yangling 712100, China

**Keywords:** *Phoma matteucciicola*, mycovirus, ourmiavirus, *Narnaviridae*

## Abstract

Here, we report a novel (+) ssRNA mycovirus, Phoma matteucciicola ourmia-like virus 1 (PmOLV1), isolated from *Phoma matteucciicola* strain LG915-1. The genome of PmOLV1 was 2603 nucleotides long and contained a single open reading frame (ORF), which could be translated into a product of RNA-dependent RNA polymerase (RdRp) by both standard and mitochondrial genetic codons. Cellular fractionation assay indicated that PmOLV1 RNAs are likely more enriched in mitochondria than in cytoplasm. Phylogenetic analysis indicated that PmOLV1 is a new member of the genus *Penoulivirus* (recently proposed) within the family *Botourmiaviridae*.

## 1. Introduction

Mycoviruses (fungal viruses) infect and replicate in all major filamentous fungal groups, yeasts, and oomycetes [1,2,3]. In contrast to known plant or animal viruses, mycoviruses usually do not cause any associated symptoms and sometimes even have beneficial effects on their fungal host [4], such as increasing heat tolerance in plant and host fungi [5] and enhancing competitiveness by producing killer proteins in yeasts [6]. However, some mycoviruses cause obvious phenotypic alterations including hypovirulence and debilitation, and can be used for biological control of fungal diseases, such as Cryphonectria hypovirus 1 (CHV1) and Sclerotiorum sclerotiorum hypovirulence-associated DNA virus 1 (SsHADV-1) against *Cryphonectria parasitica* and *Sclerotiorum sclerotiorum*, respectively [7,8].

So far, more than 300 mycoviral sequences have been reported in the National Center of Biotechnology Information (NCBI) database. The identification of newly isolated mycoviruses may contribute to our understanding of the diversity, evolution, and ecology of viruses [9,10]. Most mycoviruses have double-stranded RNA (dsRNA) or positive single-stranded RNA (+ssRNA) genomes [11,12]. Previous studies showed that ssRNA mycoviruses are classified into seven families, including *Alphaflexiviridae*, *Gammaflexiviridae*, *Barnaviridae*, *Hypoviridae*, *Mymonaviridae*, *Narnaviridae,* and *Botourmiaviridae* [11,12,13].

Viruses in the family *Narnaviridae* are the simplest viruses and have been discovered in filamentous fungi, yeast, and oomycetes [13]. The genome of this family consists of a (+)ssRNA, 2.3–3.6 kb in size, and contains a single open reading frame (ORF) that encodes an RNA-dependent RNA polymerase (RdRp). Narnaviruses usually lack coat protein (CP) and movement protein (MP), so the genomes exist in the form of an RNA/RdRp nucleoprotein complex in the lipid vesicles of cells [14]. Phylogenetically, narnaviruses are the closest relative to plant ourmiaviruses, whose genome consists of three ssRNA segments, encoding RdRp, CP, and MP, respectively [15]. Therefore, it was proposed that *Ourmiavirus* represents a link between fungal viruses and viruses of other higher organisms [14]. The family *Botourmiaviridae* currently comprised four approved genera (*Botoulivirus*, *Ourmiavirus*, *Scleroulivirus*, and *Magoulivirus*) and one recently proposed genus (*Penoulivirus*), among which *Botoulivirus*, *Penoulivirus*, and *Magoulivirus* infect only fungi; *Scleroulivirus* infects both fungi and plants; and *Ourmiavirus* comprises only viruses isolated from a plant host (https://talk.ictvonline.org/taxonomy/).

*Phoma matteuccicola* is the causal agent of leaf blight disease in *Curcuma wenyujin*, and can generate huge economic losses [16]. So far, only one virus, PmPV1, has been reported in *P. matteuccicola* [17]. In the present study, we reported a novel ourmia-like mycovirus from *P. matteuccicola* and provisionally named it Phoma matteucciicola ourmia-like virus 1 (PmOLV1). We showed here that PmOLV1 is a new member of the recent proposed genus *Penoulivirus* within the family *Botourmiaviridae*, and is likely enriched in mitochondria.

## 2. Materials and Methods

### 2.1. Fungal Isolates and Culture Conditions

*P. matteuccicola* strain LG915-1 was isolated from *C. wenyujin*, which showed symptoms of leaf blight disease in Hainan province, China, in 2016. It was cultured on potato dextrose agar (PDA) at 25 °C. For dsRNA extraction, mycelium was cultured on a PDA plate covered with cellophane membranes at 25 °C for 10 days.

### 2.2. Extraction of dsRNA

DsRNA was extracted from 3.0 g of frozen fungal mycelium using the method described previously by Morris and Dodds (1979) with minor modifications, and by selective absorption to columns of cellulose powder CF-11 (Sigma-Aldrich, St. Louis, MO, USA) in the presence of 16% ethanol [18]. To improve the yield of dsRNA, nucleic acid co-precipitator was added before precipitating RNA with isopropanol. Subsequently, the crude dsRNA was treated with RNase-free DNase I and S1 nuclease (Takara, Dalian, China).

### 2.3. cDNA Cloning and Sequencing

The purified dsRNA sample was used as a template for cDNA synthesis. The cDNA library was constructed using the NEBNext^®^Ultra^TM^ RNA Library Prep Kit (Illumina, Beijing, China) according to the methods described previously [9]. To get initial sequence clones from the dsRNA, a random primer (5′-CGATCGATCATGATGCAATGCNNNNNN-3′) targeting the cDNA was used for RT-PCR amplification. The obtained PCR products were cloned to pMD18-T vector (Takara, Dalian, China) for Sanger sequencing. The internal gaps between the initial sequences were filled by RT-PCR using sequence-specific primers designed from the obtained sequences. To clone the 5′ and 3′ terminal sequence of the dsRNA, rapid amplification of cDNA ends (RACE) was conducted as previously described [9]. Each base was identified by sequencing in both orientations from at least three separate overlapping clones.

### 2.4. Sequence and Phylogenetic Analysis

Open reading frames (ORFs) and conserved domain(s) were determined using ORF Finder program and CD-search in the National Center for Biotechnology Information (NCBI) (http://www.ncbi.nlm.nih.gov). Motifs searches were conducted in the PROSITE database (http://www.expasy.ch/). Multiple sequence alignments were performed using the CLUSTALX 1.81 program [19]. Phylogenetic tree was constructed with the maximum-likelihood (ML) method in the Molecular Evolutionary Genetics Analysis (MEGA) software version 7.0 programs with 1000 bootstrap replicates [20]. Potential secondary structures at the terminal sequences of the viral genome sequence were predicted using Mfold RNA structure software [21].

### 2.5. Isolation of Mitochondria and Western Blot

To determine the subcellular distribution of PmOLV1 mRNAs in fungal cells, an enriched mitochondria fraction was extracted using the mitochondria isolation kit (Sigma-Aldrich, St. Louis, MO, USA) according to the manufacturer’s recommendation. Then, the protein concentration of supernatant and mitochondrial enriched precipitation fraction was quantified via bicinchoninic acid (BCA). The proteins were separated using sodium dodecyl sulfate polyacrylamide gel electrophoresis (SDS-PAGE) and transferred onto polyvinylidene fluoride (PVDF) membranes. After blocked with 5% dry milk in TBST (*W*/*V*), the membranes were immunoblotted with the two antibodies: cytosolic Pgk1 (Anti-PgK1, Abcam, ab113687) and mitochondrial Porin (Anti-Porin, Abcam, ab110326). Then, the membranes were further incubated with peroxidase conjugated secondary antibodies, followed by the protein bands detected using enhanced chemiluminescence HRP substrate and images taken using Chemiluminescence instrument.

### 2.6. RNA Extraction and Quantitative Reverse Transcription PCR Analysis

RNA was extracted from 200 μL of supernatant and precipitation, respectively, using the Eastep Super Total RNA Extraction Kit (Promega, Shanghai, China). The RNA concentration was detected and the cDNA was prepared from 1 μL RNA using the RevertAid First Strand cDNA Synthesis Kit for qPCR (Thermo, MA, USA). Phoma matteuciicola partitivirus 1 (PmPV1), a co-infecting and cytoplasmic virus, was used as a control in this experiment [17]. The specific primers were designed based on the RdRp of PmOLV1 using Beacon Designer V7.92. Real-time quantitative reverse transcription PCR (RT-qPCR) was conducted to use q-PCR SYBR Green mix on a CFX manager system (Qiagen, Hilden, Germany) in 20 μL reaction. Each reaction contains 1 μL template, 10 μL master mix, and 1 μL of each primer. PCR reactions were conducted to use the follow conditions: 10 min at 95 °C, 40 cycles of 58 s at 95 °C, 30 s at 56 °C, and 10 s at 72 °C.

## 3. Results

### 3.1. Discovery and Molecular Characterization of PmOLV1

The full-length cDNA sequence of Phoma matteucciicola ourmia-like virus 1 (PmOLV1) was 2603 bp in length, with a G+C content of 53.67%. The cDNA sequence of PmOLV1 was deposited in GenBank under the accession number MN473199. It contains a single ORF being 2258 bp long that initiated at the nucleotide position 263 and terminated at position 2521, potentially encoding 751 amino acid (aa) residues with a calculated molecular weight of 83.5 kDa between the initiation AUG triplet and the termination UGA triplet (Figure 1A). In addition, the genome of PmOLV1 had a 5′-untranslated region (UTR) of 262 bp and a 3′-UTR of 82 bp (Figure 1A). Using the Mfold program, the complex secondary structures of PmOLV1 were predicted, with initial ΔG values of −8.80 kcal/mol and −12.10 kcal/mol, respectively (Figure 1B). These predictions of RNA secondary structures in PmOLV1 are a typical feature of members of the genus *Mitovirus* [22].

A homology search with BLASTP showed that this 83.5 kDa protein was most closely related to the RNA dependent RNA polymerases (RdRps) of Epicoccum nigrum ourmia-like virus 2, Penicillium sumatrense ourmia-like virus 1, and Pyricularia oryzae ourmia-like virus 1 (Table 1). Furthermore, a conserved domain database (CDD) search and multiple protein alignment confirmed that this protein had eight conserved motifs that are characteristic of the RdRps of (+) ssRNA mycoviruses (Figure 2) [12]. However, if the mitochondrial genetic code is applied, the predicted RdRp could be 16 aa longer and result in an 84.90 kDa protein between the initiation AUG triplet and the termination UAA triplet (Figure 1A).

### 3.2. Phylogenetic Analysis of PmOLV1

To analyze the relationship between PmOLV1 and other mycoviruses, phylogenetic tree was constructed based on the RdRp aa sequence of PmOLV1 and other related viral sequences, including those of the ourmia-like viruses, mitoviruses, narnaviruses, and ourmiaviruses. In the obtained phylogenetic tree, PmOLV1 was grouped with penouliviruses, such as the Phomopsis longicolla RNA virus 1, Epicoccum nigrum ourmia-like virus 2, Penicillium sumatrense ourmia-like virus 1, and Pyricularia oryzae ourmia-like virus 1 (Figure 3). Therefore, we concluded that PmOLV1 was a new member of the recently proposed genus of *Penoulivirus* within the family *Botourmiaviridae* [23].

### 3.3. Presence of PmOLV1 in Mitochondria

As the PmOLV1 genome can encode putative RdRp protein with both standard and mitochondrial codons, we further tested whether PmOLV1 could localize in mitochondria, for which we performed an enrichment experiment with the strain containing both PmPV1 (a putative cytoplasmic virus) and PmOLV1. With a commercial mitochondria isolation kit (Sigma-Aldrich, St. Louis, MO, USA) demonstrated to be efficient for isolating yeast mitochondrial fraction [24], we obtained both cellular supernatant and mitochondrial enriched precipitation fractions. On the basis of Western blot, we showed that the two fractions were enriched with the cytoplasmic marker Pgk1 and the mitochondrial marker Porin, respectively (Figure 4A). This result demonstrated an efficient subcellular enrichment with the kit. qRT-PCR was further performed with RNAs isolated from the two subcellular fractions.

The results showed that PmPV1 prevailed in the cytoplasm fraction over the mitochondrial fraction by almost 2^7^ times, as the threshold cycle (C_T_) was 16.20 ± 0.44 in the cytoplasmic fraction and 22.87 ± 0.89 in the mitochondrial (from three independent experiments) (Figure 4B). However, the PmOLV1 transcript accumulation level in the mitochondrial fraction was 2^8^ times greater than in cytoplasm, as the C_T_ was 16.52 ± 0.25 in the mitochondrial fraction and 24.49 ± 0.53 in the cytoplasmic fraction.

## 4. Discussion

In this study, we identified a novel mycovirus, PmOLV1, being isolated from the plant pathogenic fungus *P. matteucciicola*, the causal agent of leaf blight disease in *Curcuma wenyujin.* The genome of PmOLV1 contained a single open reading frame (ORF), which encoded a single protein sharing amino acid sequence identity with the RdRp sequences of ourmia-like mycoviruses. Interestingly, the ORF of PmOLV1 could be translated by both standard and mitochondrial genetic codons into a product of RdRp with UGA and UAA as termination condons, respectively. Similar translation machinery has been reported with Phomopsis longicolla RNA virus 1 (PlRV1) and Tuber aestivum mitovirus (TaMV), but their termination codons are UAG and UAA. It is worth noting that the termination code of PmOLV1 is UGA, not UAG, when standard genetic code was invoked. However, the mitoviruses always have the UAA termination codon, such as Cryphonectria parasitica mitovirus 1 (CpMV1), Sclerotinia sclerotiorum mitovirus 1 (SsMV1), and Botrytis cinereal mitovirus 1 (BcMV1). In addition, the narnaviruses always have the UGA termination codon.

Although the cellular fractionation assay indicated that PmOLV1 mRNA is likely more enriched in mitochondria than in cytoplasm, in this study, we cannot rule out the potential contamination of the mitochondrial fraction by vesicular bodies. This is only a biochemical enrichment experiment. Therefore, in the near future, additional studies are necessary in vivo to confirm whether the virus PmOLV1 is associated with mitochondria themselves or other vesicular bodies potentially also enriched in the fraction.

PmOLV1 infection is very stable in the mycelia of *P. matteucciicola* strain LG915-1. 24 strains were obtained by protoplast regeneration, 44 strains were obtained through single spore isolation, and 56 isolates were obtained through hyphal tip sub-culturing following ribavirin and cycloheximide treatment, as reported previously [25]. However, all such attempts at eliminating PmOLV1 from the mycelial cells failed (data not shown).

Phylogenetic analysis with RdRp sequence placed PmOLV1 in a distinctive clade with closely related to the ourmiaviruses rather than to any of the narnaviruses (Figure 3). It is reported that ourmiaviruses are plant viruses with three RNA genomes, which encoded an RdRp, CP, and MP, respectively [26]. Previous studies proposed that plant ourmiaviruses might have evolved from at least two or three distinct sources [27]. The RdRp genes of ourmiaviruses are closely related to the narnaviruses, while the MP gene might have been acquired from tombusviruses, and it seems likely that the CP gene was acquired from the same or a different virus [15]. However, because most mycoviruses could be transmitted via hyphal anastomosis with cytoplasmic flow, the MP seemed to be unnecessary for the life cycle of mycoviruses. Even some mycoviruses lack the coding sequence of CP. Furthermore, previous study showed that the ourmia-like viruses might have evolved from ourmiavirus that infects the plant host, and later adapted to the fungal host via losing the dispensable MP and CP [28]. In the present study, the observed sequence conservation and phylogenetic analysis strongly indicated that PmOLV1 was a new member of the recently proposed genus of *Penoulivirus* within the family *Botourmiaviridae*.

## Figures and Tables

**Figure 1 viruses-12-00231-f001:**
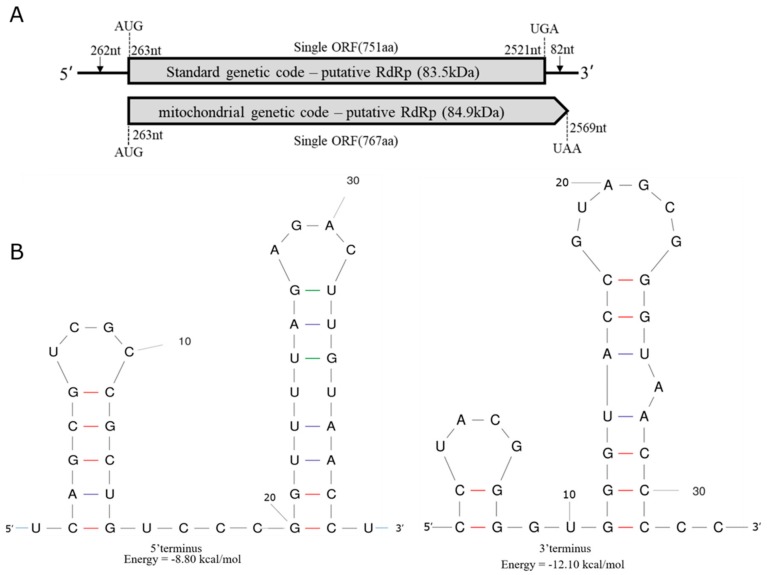
The genome organization of Phoma matteucciicola ourmia-like virus 1 (PmOLV1). (**A**) Schematic representation of PmOLV1 RNA genome. The open reading frame (ORF) and the untranslated regions (UTRs) are indicated by an open bar and single lines, respectively. (**B**) Predicted secondary structures of 5′ (**left**) and 3′ (**right**) termini of PmOLV1.

**Figure 2 viruses-12-00231-f002:**
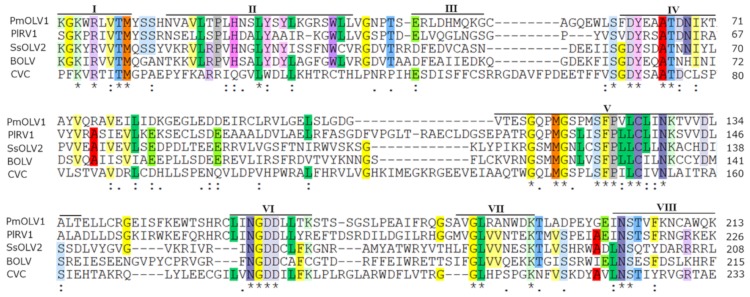
Multiple alignment of amino acid sequences of RNA-dependent RNA polymerase (RdRp) proteins encoded by PmOLV1 and other selected viruses. Horizontal lines above the alignment indicate the eight motifs by roman numerals Ⅰ to Ⅷ. Abbreviations: PlRV1, Phomopsis longicolla RNA virus 1; SsOLV2, sclerotinia sclerotiorum ourmia-like virus 2; BOLV, botrytis ourmia-like virus 1; CVC, Cassava virus C. Asterisks, colons, and dots show the same amino acid residues, conservative, and semi conservative, respectively.

**Figure 3 viruses-12-00231-f003:**
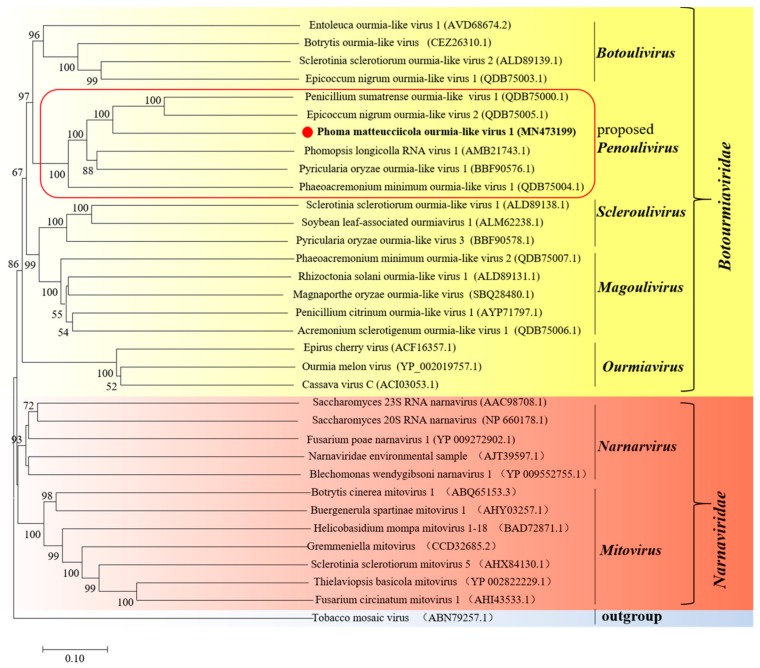
Phylogenetic analysis of PmOLV1 and other related viruses based on the deduced amino acid sequences of putative RdRps using the maximum-likelihood (ML) method with 1000 bootstrap replicates. The scale bar represents a genetic distance of 0.1 amino acid substitutions per site.

**Figure 4 viruses-12-00231-f004:**
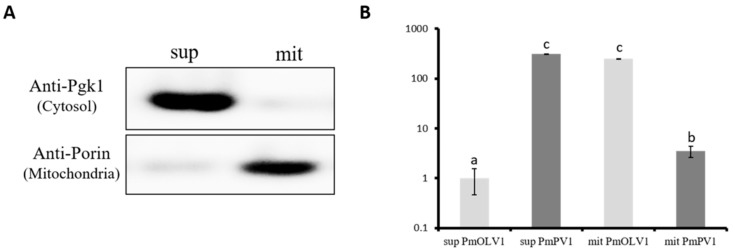
(**A**) Quality control of mitochondrial fraction isolation. The distinct fractions of supernatant (sup) and mitochondria (mit) were analyzed by immunoblotting with anti-Pgk1 (a cytosolic marker) and anti-Porin (a mitochondrial marker) antibodies, respectively, and equal amounts of protein were loaded for the blot analysis. (**B**) Comparison of PmPV1 and PmOLV1 expression levels in mitochondrial pellet (mit) and the residual supernatant (sup) from mitochondrial isolation. Values on the y-axis coordinate represented the expressional fold change relative to PmOLV1 in the supernatant fraction (sup PmOLV1) calculated with the delta delta Ct approach. Bars with different letters were statistically significant based on *t*-test at a significance level of 0.005.

**Table 1 viruses-12-00231-t001:** Information of BLASTp search results of the RNA-dependent RNA polymerase (RdRp) of Phoma matteucciicola ourmia-like virus 1.

Taxon	Virus Name	Accession	Query Cover (%)	Identity (%)	E-Value
*Penoulivirus*	Epicoccum nigrum ourmia-like virus 2	QDB75005.1	78	319/587 (54%)	0
	Penicillium sumatrense ourmia-like virus 1	QDB75000.1	83	327/605 (54%)	0
	Pyricularia oryzae ourmia-like virus 1	BBF90576.1	82	262/635 (41%)	2 × 10^138^
	Phomopsis longicolla RNA virus 1	AMB21743.1	78	245/621 (39%)	6 × 10^124^
	Phaeoacremonium minimum ourmia-like virus 1	QDB75004.1	74	219/574 (38%)	6 × 10^107^
*Scleroulivirus*	Pyricularia oryzae ourmia-like virus 3	BBF90578.1	41	74/234 (32%)	7 × 10^17^
	Sclerotinia sclerotiorum ourmia-like virus 1	ALD89138.1	55	95/405 (23%)	3 × 10^16^
	Soybean leaf-associated ourmiavirus 1	ALM62238.1	44	85/297 (29%)	6 × 10^16^
*Botoulivirus*	Epicoccum nigrum ourmia-like virus 1	QDB75003.1	49	117/429 (27%)	2 × 10^25^
	Sclerotinia sclerotiorum ourmia-like virus 2	ALD89139.1	38	79/260 (30%)	2 × 10^23^
	Botrytis ourmia-like virus	CEZ26310.1	32	79/261 (30%)	7 × 10^23^
	Entoleuca ourmia-like virus 1	AVD68674.2	31	71/257 (28%)	1 × 10^16^
*Magoulivirus*	Rhizoctonia solani ourmia-like virus 1	ALD89131.1	39	80/304 (26%)	3 × 10^19^
	Magnaporthe oryzae ourmia-like virus	SBQ28480.1	40	84/345 (24%)	2 × 10^18^
	Acremonium sclerotigenum ourmia-like virus 1	QDB75006.1	46	89/344 (26%)	1× 10^17^
	Penicillium citrinum ourmia-like virus 1	AYP71797.1	40	85/324 (26%)	5 × 10^17^
	Phaeoacremonium minimum ourmia-like virus 2	QDB75007.1	27	61/214 (29%)	2 × 10^16^
*Ourmiavirus*	Epirus cherry virus	ACF16357.1	28	73/243 (30%)	5 × 10^13^
	Ourmia melon virus	YP_002019757.1	28	68/244 (28%)	2 × 10^11^
	Cassava virus C	ACI03053.1	21	48/169 (28%)	2 × 10^10^

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
