# Peer review of "Molecular Characterization of a Novel Ourmia-Like Virus Infecting Phoma matteucciicola"

_viruses, 2020, doi:10.3390/v12020231_

Round 1

Reviewer 1 Report

Before having the final acceptance for this work, authors should clarify the following point: 1- In page 2, section 2.3. Authors explained the methodology used to obtain the complete genome sequence of this virus, but not the initial step of getting the first clones! It is not clear how and from where authors started their sequence operation. I 'm saying this because, based on their methodology; (i) RT-PCR was used to amplify gaps between clones, (ii) 5'\3' for termini amplification, but the starting sequence from they get it? Please add these information in the section 2.3 and accordingly in the results section.

Author Response

Point 1: Before having the final acceptance for this work, authors should clarify the following point: 1- In page 2, section 2.3. Authors explained the methodology used to obtain the complete genome sequence of this virus, but not the initial step of getting the first clones! It is not clear how and from where authors started their sequence operation. I 'm saying this because, based on their methodology; (i) RT-PCR was used to amplify gaps between clones, (ii) 5'\3' for termini amplification, but the starting sequence from they get it? Please add these information in the section 2.3 and accordingly in the results section. 

Response 1: Thank you so much for your efforts, suggestions and comments on our manuscript! We have added the RT-PCR details for getting the start virus sequence (page 2, section 2.3).

Reviewer 2 Report

The paper by Zhou and co-authors describes the molecular characterization of a new ourmia-like mycovirus from Phoma matteucciicola. The authors then proceed to attempt at enriching mitochondrial fractions, and based on differences in concentration with a co-infecting Partitivirus they claim this ourmia-like virus is also present in mitochondria. Furthermore, the authors search transcriptome databases with the derived RdRP, and found numerous homologous sequences (ourmia-like viruses) across vrious kingdoms.

Unfortunately the only part of the paper that is acceptable is the molecular characterization, which is not proper for a full lenght manuscript in viruses. 

The other two elements of novelty (subcellular localization and presence of homologous sequences in various database) are in one case not supported experimentally, and obvious and already reported for the latter. 

In specific, the mitochondria enriched experiment lacks foundamental controls, and the fact that you find a partitivirus in what you call mitochondrial fraction, clearly shows that your mitochondrial fraction is contaminated. In particular, it is known that narnavirales (and therefore, probably ourmi-like virus) replicate in vescicular bodies, that will specifically enrich together with mitochondria. Therefore, to prove that your virus is mitochondrial you need in vivo studies at the elctron microscope, or with a confocal microscope, with specific markers for mitochondria, and a fluorescent probe for the virus. Your results simply say that Partitivirus and your ourmia-like virus concentrate in different compartments (which is expected, given the different nature of the two viruses, one encapsidated and one naked), but it does not say anything about what compartment it is in. 

Then your mining trancriptomes does not provide any new information, since already a great number of ourmia-like viruses from metagenomics or from invertebrate samples of various nature are present in NCBI databases (and you have not considered in your tree): Furthermore, finding an ourmia-like virus in a transcriptome does not guarantee that the host of the virus is the main subject of the transcriptome, because in plant transcriptomes and animal transcriptomes, a number of endophytic or ectophytic fungi are associated. 

Finally, Ourmia-like viruses have been recently placed taxonomically in the family botourmiaviridae, and in four genera (please check ICTV web page, and you will see that that is the case).

I am afraid this paper is very far from having the necessary experimental soundness to be published in Viruses. Probably you can publish the first part, in a taxonomical journal. 

Author Response

Point 1: In specific, the mitochondria enriched experiment lacks foundamental controls, and the fact that you find a partitivirus in what you call mitochondrial fraction, clearly shows that your mitochondrial fraction is contaminated. In particular, it is known that narnavirales (and therefore, probably ourmi-like virus) replicate in vescicular bodies, that will specifically enrich together with mitochondria. Therefore, to prove that your virus is mitochondrial you need in vivo studies at the elctron microscope, or with a confocal microscope, with specific markers for mitochondria, and a fluorescent probe for the virus. Your results simply say that Partitivirus and your ourmia-like virus concentrate in different compartments (which is expected, given the different nature of the two viruses, one encapsidated and one naked), but it does not say anything about what compartment it is in. 

Response 1: Thanks for your comments. We have added the western blot experiment to detect the quality of the isolated mitochondria. As for the enrichment experiment, we demonstrated that the ‘supernatant’ and the ‘mitochondrial’ fractions were enriched with the cytoplasmic and mitochondrial markers, respectively. This data gave us some confidence regarding the mitochondrial enrichment efficiency of the kit.

    Still, we cannot rule out the potential contamination of the mitochondrial fraction by vesicular bodies. However, several lines of evidences indicate that PmOLV1 is more likely to be enriched in mitochondria than in vesicular bodies. First, the virus can use both standard and mitochondrial codons. Second, the genome of mycovirus being localized in vesicular bodies generally contains a proline-rich region (PRR) (Wu et al., 2012), which is not the case for PmOLV1. Third, previous studies showed that viruses in the family Narnaviridae replicate in the cytoplasm but not in vesicular body (Ohkita et al., 2019; Solorzano et al., 2000).

    Indeed, this is a biochemical enrichment experiment, data based on in vivo experiment would definitely be more straight forward. We would like to do so, however, it is difficult for our laboratory to do the experiment according to the existing technology and conditions, especially in a short period.

Point 2:Then your mining trancriptomes does not provide any new information, since already a great number of ourmia-like viruses from metagenomics or from invertebrate samples of various nature are present in NCBI databases (and you have not considered in your tree): Furthermore, finding an ourmia-like virus in a transcriptome does not guarantee that the host of the virus is the main subject of the transcriptome, because in plant transcriptomes and animal transcriptomes, a number of endophytic or ectophytic fungi are associated. Finally, Ourmia-like viruses have been recently placed taxonomically in the family botourmiaviridae, and in four genera (please check ICTV web page, and you will see that that is the case).

Response 2: We sincerely thank you for your good advice! We have deleted the content related to this part of analysis in the article.

    Thank you for your reminder regarding the taxonomic status of the family Botourmiaviridae. We have rebuilt the phylogenetic tree and showed that PmOLV1 is a new member of the genus Penoulivirus (recently proposed) within the family Botourmiaviridae.

Reviewer 3 Report

In this manuscript, Jia Zhou et al. found a novel ourmia-like virus infecting Phoma matteucciicola, a plant pathogenic fungi, and performed molecular characterization of the virus. This article is well written and easy to read, but I think this report is not so different from kinds of sequence annotation report. Although various virus sequences are included in EST and TSA, this information is insufficient to show scientific significance of this study because the origin is not clear as the authors have discussed. In the fractionation experiment, which could be data showing significance, I questioned both the method and conclusion. This report should not be published in this journal unless authors can provide clear and convincing explanations and data for the fractionation experiment.

Major

If there is a paper showing the application of the mitochondrial fraction purification kit to filamentous fungi, please add reference (Line 95) because I could not find it and this kit looks applicable only for animal cells. If there is not such a paper, authors have to describe principle of the mitochondrial fraction purification kit and add data which show the purity of mitochondria and cytosol fraction. In some cases, cell disruption may be insufficient. Are they eliminated in this method? Reference 17 does not show that the partitivirus is cytoplasmic one. I know that, in general, Paritiviruses are considered cytoplasmic, but if they are to be used as controls for localization experiments, they must be experimentally verified. Since the volume of mitochondria is small compared to that of the cytoplasm, the amount of RNA obtained may differ in those fractions when separated from the same sample. But, authors used same amount of RNA from supernatant and precipitation for cDNA synthesis and following experiments (Line 100). To discuss about the enrichment of PmOLV1, authors need to add experiments using different amount of RNAs originate from same amount of fungal cells. Authors should not use “concentration” (Line 178-179) because it is difficult to determine the volume of mitochondria and cytoplasm.

Minor

The information (for example, how many single spores were obtained) is valuable. Please provide specific numbers. Please add amino acid position in Fig2. What is the ordering standard in taxon in Table 1? Please sort in order of e-value, etc. Please add a reference (Line 135). Please show the threshold used for BLASTX analysis in Materials and Methods (Line 113).

Author Response

Point 1: If there is a paper showing the application of the mitochondrial fraction purification kit to filamentous fungi, please add reference (Line 95) because I could not find it and this kit looks applicable only for animal cells. If there is not such a paper, authors have to describe principle of the mitochondrial fraction purification kit and add data which show the purity of mitochondria and cytosol fraction. In some cases, cell disruption may be insufficient. Are they eliminated in this method? Reference 17 does not show that the partitivirus is cytoplasmic one. I know that, in general, Partitiviruses are considered cytoplasmic, but if they are to be used as controls for localization experiments, they must be experimentally verified. Since the volume of mitochondria is small compared to that of the cytoplasm, the amount of RNA obtained may differ in those fractions when separated from the same sample. But, authors used same amount of RNA from supernatant and precipitation for cDNA synthesis and following experiments (Line 100). To discuss about the enrichment of PmOLV1, authors need to add experiments using different amount of RNAs originate from same amount of fungal cells. Authors should not use “concentration” (Line 178-179) because it is difficult to determine the volume of mitochondria and cytoplasm.

Response 1: We sincerely thank you for your good advice! The kit has been applied for yeast (Guirola et al., 2010), the reference has been added. For better experiment control, the distinct fractions of supernatant (sup) and mitochondria (mit) were analyzed by immunoblotting with anti-Pgk1 (a cytosolic marker) and anti-Porin (a mitochondrial marker) antibodies in Figure 4A of the revised manuscript. The results showed that the mitochondria were well separated from the cytoplasm, indicating high enrichment efficiency.

    We have re-performed the qRT-PCR experiment based on the reviewer suggestion in Figure 4B. In addition, the ‘concentration’ was changed to ‘transcript accumulation level’.

Point 2:The information (for example, how many single spores were obtained) is valuable. Please provide specific numbers.

Response 2: Thanks! We have provided the specific numbers in the manuscript (Line 203-207) according to our original data.

Point 3: Please add amino acid position in Fig2.

Response 3: Thanks! The amino acid position is added in Figure 2.

Point 4: What is the ordering standard in taxon in Table 1? Please sort in order of e-value, etc.

Response 4: We have reordered it according to the e-value.

Point 5: Please add a reference (Line 135).

Response 5: Yes, we have added the related reference in line 138.

Point 6: Please show the threshold used for BLASTX analysis in Materials and Methods (Line 113).

Response 6: The threshold was presented at the bottom of the supplementary Table S1.

Round 2

Reviewer 2 Report

The authors in my opinion still failed to demonstrate that this ourmiavirus is indeed localized in mitochondria (which is the only new aspect revealed in their study).

Regarding the first point they use to support the mitochondrial localization (which is also a core point of their paper), i.e. the different protein sequence using mitochondrial and genetic codes, the same happens for example with Narnavirus 23S, a virus known to be cytoplasmic.

UNIVERSAL GENETIC CODE

>NC_004050.1 Saccharomyces 23S RNA narnavirus, complete genome_rframe1_ORF

MHHKVNVKTQREVHFPMDLLQACGASAPRPVARVSRATDLDRRYRCVLSLPEERARSVGCKWSSTRAALRRGLEELGSREFRRRLRLADDCWRAICAAVCTGRKFPSFSVTDRPARARLAKVYRMGRRLLVGVVCRGESVVSDLKQECADLRRVIFEGSTRIPSSSLWGLVGVLGWTSPERAMQLTFIGRALPYGSPDVERRALASHAATLSIPAECHPNYLVAAEQFAKSWADDNLPRKFRIYPIAVQESSCMEYSRAQGGLLQSFRKGFVGYDPAAPSADPDDLELAKERGFSRIRASWYSTFRYRGELKSTNQSLEARVAVVPERGFKARIVTTHSASRVTFGHQFRRYLLQGIRRHPALVDVIGGDHRRAVETMDGDFGLLRPDGRLLSADLTSASDRIPHDLVKAILRGIFSDPDRRPPGTSLADVFDLVLGPYHLHYPDGSEVTVRQGILMGLPTTWPLLCLIHLFWVELSDWAPARPNHSRGFVLGESFRICGDDLIAWWRPERIALYNQIAVDCGAQFSAGKHLESKTWGIFTEKVFTVKPVKMKVRVRSEPSLKGYVFSRSSAFSCRMGGKGITGIRAARLYTIGAMPRWSRRIRDVYPGSLEHRTASQRYGEPVTVYRFGRWSSAIPLRWAVRAPTRTVGNPVQSLPDWFTVGPAASSVAADSNAFGAVSRVLRRMFPGLPRKLASAGIPPYLPRVFGGGGLVKSTGLTTKIGAVASRRWMSRIGHDLYRSRERKSTLGRVWTLSTSPAYAASLHEVEKFMDRPDIILTRKCRNPMLKHARELGLFEEVFESRVGGGILWASLNGKALVESHSPSILQVSRNLRRSLACPSGGFLRPSAPIGKLVQRHTLPRGTVWFLESSATDSARQGGMGLPPPPPPPLGGGGMAGPPPPPFMGLRPESSVPTSVPFTPSMFSERLAALESLFGRPPPS

YEAST MITOCHONDRIAL

>NC_004050.1 Saccharomyces 23S RNA narnavirus, complete genome_rframe1_ORF

MHHKVNVKTQREVHFPMDTLQACGASAPRPVARVSRATDTDRRYRCVTSLPEERARSVGCKWSSTRAALRRGTEETGSREFRRRTRLADDCWRAICAAVCTGRKFPSFSVTDRPARARTAKVYRMGRRTTVGVVCRGESVVSDTKQECADLRRVIFEGSTRIPSSSLWGTVGVTGWTSPERAMQTTFIGRATPYGSPDVERRATASHAATTSIPAECHPNYTVAAEQFAKSWADDNTPRKRIYPIAVQESSCMEYSRAQGGTTQSFRKGFVGYDPAAPSADPDDTELAKERGFSRIRASWYSTFRYRGELKSTNQSTEARVAVVPERGFKARIVTTHSASRVTFGHQFRRYTTQGIRRHPATVDVIGGDHRRAVETMDGDFGTLRPDGRTTSADLTSASDRMPHDLVKAILRGIFSDPDRRPPGTSTADVFDTVTGPYHTHYPDGSEVTVRQGITMGTPTTWPTTCTIHTFWVETSDWAPARPNHSRGFVTGESFRICGDDTIAWWRPERIATYNQIAVDCGAQFSAGKHTESKTWGIFTEKVFTVKPVKMKVRVRSEPSLKGYVFSRSSAFSCRMGGKGMTGIRAARTYTIGAMPRWSRRIRDVYPGSTEHRTASQRYGEPVTVYRFGRWSSAIPTRWAVRAPTRTVGNPVQSLPDWFTVGPAASSVAADSNAFGAVSRVTRRMFPGLPRKTASAGIPPYTPRVFGGGGTVKSTGTTTKIGAVASRRWMSRIGHDTYRSRERKSTLGRVWTTSTSPAYAASTHEVEKFMDRPDIILTRKCRNPMTKHARETGLFEEVFESRVGGGITWASTNGKATVESHSPSITQVSRNTRRSTACPSGGFTRPSAPMGKTVQRHTLPRGTVWFLESSATDSARQGGMGTPPPPPPPTGGGGMAGPPPPPFMGTRPESSVPTSVPFTPSMFSERTAALESLFGRPPPSwwqngetrsaqarynyrawa

The second argument they use, is their cell fractionation protocol to enrich mitochondria: 

The western markers they choose are not sufficient to demonstrate that what they call mitochondrial fraction is rich of other components of the microsomal fraction (vesicle and other membrane bound proteins or other cytoplasmic compartments). In facts, they used as control for cytoplasm a soluble protein (PgK1, a glycolysis protein); therefore, it is not a good control for microsomal fraction in the cytoplasm. In other words, they choose a soluble protein, that is only a marker for the cytosol, but we all know the cytoplasm is made of a number of membranous organelles that can still be in the mitochondrial fraction.

Furthermore the Wu et al. 2012 paper they quote to support the idea that to be membrane bound (or vesicle bound) a proline-alanine rich region is needed, is again a generalization that is not supported by many other works: the proline alanine rich motif identified in that paper is probably necessary to target some specific subset of membranes (in the discussion they refer to insect cells) related only to the specific viruses they mention, but most RNA virus RdRP interact with membranes without any specific proline-alanine motif.

I have used the Sigma kit quoted to purify “mitochondria”, and I can assure, that other membrane-bound molecular markers co-purify with the supposed “mitochondria”. If you want to demonstrate you have only mitochondria in that fraction, you need to use markers of other compartments, golgi, ER, vacuoles, etc...

Finally, the Ohkita  et al paper  2019 the authors mention does not provide any direct evidence of what they claim, and Ohkita do not discuss “vesicular” localization: in the second paragraph of the discussion they simply say that since the virus is targeted by silencing defence, it is indeed cytoplasmic (and not mitochondrial) (and I agree with them with this conclusion). But I can give you an example of a virus, CHV1, which is vesicular, and is indeed targeted by anti-viral silencing. Therefore you can still be membrane-bound, and be a target of silencing.

Author Response

Thanks for your comments!

We have converted our manuscript into a Communication instead of an Article and discuss our findings on virus localisation according to your suggestions and those of the journal editor board.

Reviewer 3 Report

I have no comment.

Author Response

We sincerely thanks for your comments!